# Biosensors Based on Ion-Sensitive Field-Effect Transistors for HLA and MICA Antibody Detection in Kidney Transplantation

**DOI:** 10.3390/molecules27196697

**Published:** 2022-10-08

**Authors:** Thu Zar Ma Ma Moe Min, Sonwit Phanabamrung, Woraphan Chaisriratanakul, Apirak Pankiew, Awirut Srisuwan, Kondee Chauyrod, Cholatip Pongskul, Chamras Promptmas, Chanvit Leelayuwat

**Affiliations:** 1Department of Medical Laboratory Technology, University of Medical Technology, Mandalay, Patheingyi 05071, Myanmar; 2Department of Clinical Immunology and Transfusion Sciences, Faculty of Associated Medical Sciences, Khon Kaen University, Khon Kaen 40002, Thailand; 3The Centre for Research and Development of Medical Diagnostic Laboratories, Faculty of Associated Medical Sciences, Khon Kaen University, Khon Kaen 40002, Thailand; 4Thai Microelectronic Center (TMEC), National Science and Technology Development Agency, Chachoengsao 24000, Thailand; 5Division of Nephrology, Department of Medicine Khon Kaen University, Khon Kaen 40002, Thailand; 6Department of Biomedical Engineering, Faculty of Engineering, Mahidol University, Nakhon Pathom 73170, Thailand

**Keywords:** antibody detection, ISFET immunosensor, protein immobilization, silicon nitride, HLA, MICA

## Abstract

This work demonstrates the ability of the Ion-Sensitive Field-Effect Transistor (ISFET)-based immunosensor to detect antibodies against the human leukocyte antigen (HLA) and the major histocompatibility complex class-I-related chain A (MICA). The sensing membrane of the ISFET devices was modified and functionalized using an APTES-GA strategy. Surface properties, including wettability, surface thickness, and surface topology, were assessed in each module of the modification process. The optimal concentrations of HLA and MICA proteins for the immobilization were 10 and 50 μg/mL. The dose-response curve showed a detection range of 1.98–40 µg/mL for anti-HLA and 5.17–40 µg/mL for anti-MICA. The analytical precision (%CV) was found to be 10.69% and 8.92% for anti-HLA and -MICA, respectively. Moreover, the electrical signal obtained from the irrelevant antibody was considerably different from that of the specific antibodies, indicating the specific binding of the relevant antibodies without noise interference. The sensitivity and specificity in the experimental setting were established for both antibodies (anti-HLA: sensitivity = 80.00%, specificity = 86.36%; anti-MICA: sensitivity = 86.67%, specificity = 88.89%). Our data reveal the potential of applying the ISFET-based immunosensor to the detection of relevant anti-HLA and -MICA antibodies, especially in the field of kidney transplantation.

## 1. Introduction

Kidney transplantation (KT) is the best treatment for patients who suffer from an end-stage renal disease, providing benefits in terms of both survival and quality of life compared with maintenance dialysis therapy. However, antibodies against the human leukocyte antigen (HLA) and the major histocompatibility complex class I-chain-related gene A (MICA) that are pre-formed or de novo produced in the recipient’s circulation can cause kidney damage and dysfunction [1,2,3]. Due to the extensive polymorphism of both HLA and MICA proteins, antibody screening is necessary and routinely performed in pre- and post-KT patients for the diagnosis of kidney rejection due to antibody-mediated mechanisms. Several technologies have been developed for anti-HLA and -MICA detection, such as ELISA [4]. and Luminex^®®^ [5]. Although the bead-based assay utilizing the Luminex fluorochrome instrument has been established as the “gold standard” for HLA and MICA antibody testing, it has some drawbacks, such as the presence of denatured proteins on the beads, which reveal cryptic epitopes, and the issue of obtaining appropriate fluorescence cut-off values for positivity [6]. Furthermore, there are some limitations to antibody identification for broadly sensitized patients. Other limitations include the high cost of the reagent kits, the requirements for the well-educated analysis and interpretation of data in the critical case, and its technically demanding nature, which make it inappropriate for use in resource-limited countries.

The ion-sensitive field-effect transistor (ISFET) is one of the most appealing electrochemical biosensors, and is currently used for biomolecular detection [7,8,9,10,11]. It has several favorable characteristics such as high sensitivity, high specificity, low detection limit, rapid and real-time detection, as well as an economical price, and miniaturization abilities. Therein, biomolecule immobilization, types of sensing material, and types of immobilization methods are the key factors that influence the performance of the developed biosensor. Among the many types of sensing materials, silicon nitride (Si3N4) is the most widely used, as it possesses high electroconductivity, good mechanical stability, and low intrinsic fluorescence [12]. However, the immobilization of biomolecules on silicon surfaces is still challenging due to the electrically neutral and non-porous properties of these materials. To overcome these problems, several surface modification strategies were introduced. One of them, covalent immobilization, is the most preferable method due to the strong linkage between the biomolecule and the functional group on the silicon substrates [13]. Firstly, surface activation was performed to generate the –OH group on the silicon substrate. Wet cleaning using strong oxidizing solutions (piranha, RCA-1, and hydrofluoric acid) and oxygen plasma treatment is often used for this step [7,11,12,13,14,15]. The strong oxidizing agents and the oxygen plasma detach the oxides and the organic contaminants and produce more hydrophilic properties on the surface by forming the hydroxyl (–OH) group. Secondly, the formation of the –OH group enhances the functionalization steps using 3-aminopropyl-triethoxysilane (APTES) and glutaraldehyde (GA) [16,17]. It is important to form the monolayer of the APTES molecule on the surface because the multilayer-APTES is fragile and easily removed during this process. Pawasuth, et al. reported the optimal concentration of APTES and incubation time to obtain a thin and stable APTES layer for immense biomolecules’ immobilization, at 1% APTES, and 1 h, respectively. This condition was proven to increase the ISFET sensitivity in electrical measurements [7]. Nevertheless, the immobilized protein may not be stable on the APTES-modified surface because there are weak interactions between the –NH_2_ group of APTES and the biomolecules [18]. For this reason, cross-linking with GA was also studied. GA possesses the bifunctional aldehyde groups that can build a covalent bond with the –NH_2_ group of both APTES-modified surface and biomolecules [12,17,19]. Another problem in the development of immunosensors using the APTES-GA strategy is controlling the complexity of protein immobilization and antigen-antibody interactions on the modified surface. Moreover, Debye screening length (λD) plays a critical role in determining the sensing performance of FET-based biosensing measurements. This is a physical distance where the charged analytes are electrically screened by ions dissolved in the medium [20,21]. It is apparent that the ISFET sensing device can only detect the changes in charge density that occur within the Debye length. The greater the distance of the charges from the sensing surface, the lower the sensitivity of ISFET is. In the case of macromolecules (generally greater than 10 nm) that are longer than the Debye screening length of the physiological salt solution, the protein charges will be screened out beyond the Debye length [20]. The Debye lengths at different concentrations of the phosphate buffer saline (PBS) at room temperature (RT) are 0.7, 2.3, 7.3 and 23 nm in 1×, 0.1×, 0.01× and 0.001× PBS, respectively [21]. Thus, it is essential to optimize all the above factors for each protein immobilization to ensure efficient detection by ISFET. 

In the present work, we mainly focused on the development of ISFET biosensors for anti-HLA and -MICA antibody detection using the APTES-GA strategy in the protein immobilization step. To investigate the surface properties after APTES-GA functionalization, several techniques were used. The surface wettability, surface thickness, and surface topology were examined using contact angle (CA), ellipsometry, and atomic force microscopy (AFM), respectively. The existence of immobilized-HLA and -MICA on the modified Si_3_N_4_ surface was studied by probing with the FITC-labelled antibody. Then, the fluorescent signal was visualized by a fluorescence microscope. ISFET measurements and method validation were also conducted to establish the prototype and indicate the application of this immunobiosensor as an alternative method in HLA and MICA antibody detection, especially for cases of kidney transplantation.

## 2. Results and Discussions 

### 2.1. Contact Angle (CA) Measurement 

The CA was widely used to monitor the surface tension or wettability of solid surfaces. As shown in Figure 1, the CA value of the bare ISFET surface was 47.05 ± 2.48°, which was slightly hydrophobic. After cleaning this, and activating it in the RCA-1 solution, the CA value was 30.88 ± 6.49°, indicating a change in the surface properties, from hydrophobic to hydrophilic. This may be due to the removal of organic contaminants and the formation of -OH groups and/or Si-OH groups on the treated silicon substrates. The hydrophobicity property was shown on the APTES-modified Si_3_N_4_ surface with a CA of 66.44 ± 3.58° due to the substitution of the hydroxyl group with the alkyl chain and amine functional group from the APTES molecules. The hydrophilicity of silicon substrate tends to increase after GA functionalization, as observed by the decrease in the CA to 55.76 ± 8.15° when compared to the APTES-treated surface. This result is consistent with the value reported by previous studies [12,16,20]. After protein immobilization, the CA was found to be 78.37 ± 4.04°, which was attributed to the amine group found on the protein structure.

### 2.2. Surface Thickness

The thin film thickness of the modified Si_3_N_4_ surfaces in each step was determined using a spectroscopic ellipsometer (Table 1). The reported value was averaged from three different samples (*n* = 3) on the same surface with standard deviation (SD). The thicknesses of the APTES layer and GA layer were 2.51 ± 0.17 nm and 1.52 ± 0.46 nm, respectively. These findings are almost equivalent to the changes in surface thickness after silanization using APTES and functionalization by GA [16]. Evidently, using 1% APTES for 1 h can provide the APTES monolayer (approximately 2 nm) on the Si_3_N_4_ surface with high sensitivity to electrochemical measurement [7,20,21]. The results from our study revealed the APTES chains’ attachment to hydroxyl (-OH) groups on the activated surface with little oligomerization. 

### 2.3. Surface Topography

The topographic AFM images of functionalization and protein immobilization on Si_3_N_4_ surfaces are shown in Figure 2. The root mean square (RMS) values were calculated as the average values, measured from three different positions on the surface of the samples. The mean surface roughness of 2.09 ± 1.22 nm was obtained on a bare Si_3_N_4_ surface (Figure 2a). The RMS roughness considerably decreases to 0.41 ± 0.14 nm after surface activation with the RCA-1 solution (Figure 2b), indicating the removal of organic contaminants from the bare surface. There are various ways of grafting the APTES layer, which critically affects the stability of APTES and the sensitivity of the biosensors. During the silanization process, multilayer APTES can be found, due to the presence of water molecules. This may lead to a fragile and non-uniform surface. To avoid this phenomenon, it has been suggested that the silicon substrate is cured at a high temperature to remove the water molecules before silanization and stop the polymerization of APTES after silanization [16]. Moreover, non-aqueous solvents such as toluene, ethanol, and methanol were used to prepare the APTES solution [7,16,19]. In this study, 1% APTES in 90% ethanol and 2.5% GA were employed on the silicon surface. The RMS value increases to approximately 1.46 ± 0.42 nm on APTES-modified surfaces compared to those which have undergone surface activation, indicating the formation of a silane layer on the surface (Figure 2c). However, compared with the native Si_3_N_4_ surface, the RMS value of the APTES-modified surface moderately decreased. These data imply that the grafting of the APTES molecules was very uniform on the silicon substrate. This finding was consistent with the previous study [22]. The APTES-modified Si_3_N_4_ surface after functionalization with 2.5% GA increased the RMS roughness to approximately 2.80 ± 1.51 nm. This indicates a few aggregations of GA on the APTES-modified surface (Figure 2d). In addition, the AFM images of the protein-immobilized surface showed several populated fine peaks on the modified Si_3_N_4_ substrates with increasing RMS values of 6.55 ± 2.2 and 3.46 ± 1.51 in HLA and MICA protein immobilization, respectively (Figure 2e,f). This result suggested that the HLA and MICA proteins were successfully immobilized on the functionalized silicon surface. 

### 2.4. Fluorescence Detection

The HLA-A and MICA*010 proteins were used to denote the accomplishment of the immobilization procedure against their corresponding antibodies. In this approach, the isotype-matched control antibody was used as a negative control. The concentrations of immobilized HLA-A and MICA*010 proteins were 1, 2.5, 5, 10, and 20 µg/mL. The integrated fluorescence densities of both HLA-A and MICA*010 were increased according to the concentration of immobilized proteins and reached a maximum of 10 µg/mL (Figure 3). However, a tiny amount of fluorescence pixel intensity occurred on the binding of the isotype control surface due to the nonspecific binding reactions between the FITC-tagged antibody and the residual reactive aldehyde group on the functionalized surface. These results imply that the modified silicon substrate was appropriate for the HLA and MICA proteins’ immobilization, preserving the antibody-binding property of the immobilized proteins. 

### 2.5. ISFET Measurement

To explore the optimal current that is applied as a constant current to the system, I_ds_-V_gs_ curves in the ISFETs were measured in 10 µM PBS, pH 7.4 at RT using the I_ds_-V_gs_ characterization semiconductor analyzer, B1500A. As shown in Figure 4a, a manifest shift in the I_ds_-V_gs_ curves was found at each measurement step. The linear and parallel regions of the responsive voltages (V_gs_) were observed in the I_ds_ range between 20 µA and 40 µA (Figure 4b). Therefore, the constant current of 30 µA was selected as the optimal current for the electrical measurement. The V_gs_ largely decreased after silanization and cross-linking with GA (V_gs (GA)_). This phenomenon can be explained by the positive charges of the carbonyl carbon in the GA molecule. The positive charges on the sensing membrane induced a reduction in the gate voltage. After protein immobilization, the positive charges were replaced by the negative charges of immobilized proteins owing to the increasing V_gs_
_(Protein__)_. 

The changes in voltage drift error (V_df_) were investigated in bare, HLA-A, and MICA*010-immobilized ISFETs at 1 min intervals up to the 10 min mark in PBS, pH 7.4, at RT. As shown in Figure 5, the drift error continuously increased in the bare ISFET. However, this value was reduced after HLA-A and MICA immobilization. The drift error reached the stable point after 5 min, as the immobilized proteins prevented the random reactivity of unwanted ions in PBS buffer to residual chemical links on the insulated gate [23]. 

The sensitivity of ISFET measurement is influenced by the Debye screening length, which is determined by the ionic strength of the measuring buffer. In this experiment, the 3D structure of HLA (PDB code: 3BO8) and MICA (PDB code: 1B3J) proteins were retrieved from the protein databank (https://www.rcsb.org, accessed on 1 May 2022) to estimate the size of those proteins. The longest dimensions of HLA and MICA proteins are approximately 7.31 nm and 7.87 nm, respectively (Figure 6a,b). The total length of the grafted Si_3_N_4_ surface and immunocomplex of both proteins is about 19–22 nm (Figure 6c). Therefore, the concentration of 10 µM PBS buffer, which provides a Debye length of approximately 23 nm, was chosen for electrical measurement [24].

### 2.6. Protein Concentration Optimization for Electrical Measurement

The efficiency of the ISFET biosensor in the detection of target analytes is influenced by the density of the bioreceptor that is immobilized on the sensing membrane of the ISFET device. We performed HLA and MICA protein optimization to explore the optimal protein concentration for the immobilization step. The ISFET devices were immersed in different concentrations of protein solutions prepared in 1× PBS, pH 7.4. The magnitude of the ΔV_gs_ reflected the surface potential induced by the charged proteins that were immobilized on the gate surface. The ΔV_gs_ were increased and reached the stable point at the 10 and 50 µg/mL concentrations for HLA and MICA, respectively (Figure 7). The stable signal is the result of the steric repulsion between the immobilized proteins on the gate surface and the effect of a small functioning area (approximately 1 × 2 mm.) of the ISFET device. 

The optimal protein concentration was immobilized on the Si_3_N_4_ surface to detect the relevant antibodies. As shown in Figure 8, the gate potential changes after protein immobilization increased to around 10–20 mV, indicating the achievement of protein immobilization. A minor change in surface potential was observed after the blocking and reducing step. This could be explained by the neutral charge of the glycine, which has a small effect on the gate potential induction. Moreover, blocking with glycine helps to block the free reactive aldehyde group and reduces the interference generated by the free ions found in the measuring buffer. The gate potential was considerably increased in the specific antibodies when compared to the isotype control (Figure 8). These results suggested specific interactions between antibodies and their corresponding antibody, immobilized on the ISFET surface.

### 2.7. Method Validations

The analytical performance of the developed ISFET immunosensor for anti-HLA and MICA detection was evaluated and described in terms of the limit of detection (LoD), cut-off determination, analytical precision, sensitivity, and specificity in the experimental setting. 

#### 2.7.1. Dose–Response Curve 

The dose-response curve was established to evaluate the linearity range and sensitivity of the ISFET-based HLA and MICA sensors. The dose-response curve was illustrated between ΔV_gs_ and different concentrations of antibodies (Figure 9a,c). Antibodies against HLA and MICA were used as the specific antibodies. As the ISFET sensor rapidly responds to the charged molecules or ions in the measuring environment, it is important to consider the interference caused by the non-specific binding of the antibody to the reactive aldehyde group. To overcome this problem, the purified mouse IgG1, ĸappa-isotype antibody, which has no relevant specificity to the HLA and MICA proteins, was used to eliminate the non-specific signal. The ΔV_gs_ was gradually increased and saturated at the concentration of 40 and 80 µg/mL of anti-HLA and MICA, respectively. This result suggested the maximum binding ability of the proteins that were immobilized on the gate surface to the antibodies. Moreover, it can be assumed that the surface modification strategy used in this study efficiently provides a proper functional group for protein immobilization with a low impact on protein structure or function. However, the ΔV_gs_ obtained from the isotype antibody was substantially lower than that of the specific antibody, indicating the binding ability of the specific antibody to its corresponding antigen with minimum noise background. 

As shown in Figure 9b,d, the linear range was observed at 0–40 µg/mL of both anti-HLA and -MICA with a good correlation coefficient (R^2^ = 0.9862 and 0.9986 for anti-HLA and -MICA, respectively). The LoD of this immunosensor, calculated using the slope of the curve and the sample blank data, was determined as 1.98 µg/mL for anti-HLA and 5.17 µg/mL for anti-MICA. The LoQ was identified as 6.59 µg/mL and 17.22 µg/mL for anti-HLA and anti-MICA, respectively. The cut-off value was calculated using the ΔV_gs (antibody)_ data obtained from the isotype control at various concentrations (0–160 µg/mL). The cut-off value utilized to differentiate between the positive and negative signals was established at 4.78 and 9.12 mV for anti-HLA and anti-MICA, respectively. Moreover, the sensitivity of the ISFET device for anti-HLA and -MICA detection was defined by the slope of the curve at every antibody concentration value [24]. Therefore, the detection range of this developed immunosensor for anti-HLA is 1.98–40 µg/mL, and for anti-MICA, it is 5.17–40 µg/mL, with a sensitivity of 0.32 mV/µg for anti-HLA and 0.23 mV/µg for anti-MICA. The sensitivity obtained from this study is slightly lower than that of the previous report [11]. Since the sensitivity of the ISFET biosensor is strongly influenced by the Debye screening length, using a small antigenic peptide instead of the whole-protein immobilization is another potential approach that could allow for the immunocomplex to occur near the sensing membrane. Moreover, increasing the surface-to-volume ratio by adding nanostructures helps to improve protein immobilization and, hence, the sensitivity [19]. 

#### 2.7.2. Analytical Precision

The reproducibility of the developed biosensor was determined using ten independent HLA-ISFET and ten MICA-ISFET sensors from three different batches under the same analysis conditions. The concentration of antibodies used to assess the assay precision of both anti-HLA and -MICA was 20 µg/mL, which falls in the linearity range of the calibration curve for both anti-HLA and -MICA. The ΔV_gs_ obtained from each ISFET was converted to the antibody concentration using the calibration curve. As shown in Table 2, the coefficient of variation (%CV) was calculated to be 10.69% for anti-HLA and 8.92% for anti-MICA, suggesting that acceptable reproducibility was obtained from the proposed biosensor. The reproducibility of this study is in accordance with the previous report, which utilized the Si_3_N_4_-ISFET immunosensor to detect the urinary albumin that reported the %CV of the inter-assay to detect 100 µg/mL of the human serum albumin: 9.67% [25]. 

#### 2.7.3. Sensitivity and Specificity of the ISFET-Based Immunosensor for Anti-HLA and -MICA Detection in the Experimental Setting

The sensitivity and specificity of the ISFET-based HLA biosensor were evaluated using the commercial anti-HLA-A and household anti-MICA (WJ-1) as known positive antibodies. The purified mouse IgG1, ĸappa-isotype control antibody represented known negative antibodies. The sensitivity and specificity were calculated using a 2 × 2 table for the experimental setting. The sensitivity and specificity of the developed ISFET-based anti-HLA sensor in this experimental setting were 80.00 % and 86.36 %, respectively. Likewise, the sensitivity and specificity of the ISFET sensor for anti-MICA detection were found to be 86.67% and 89.89%, respectively. 

## 3. Materials and Methods

### 3.1. Modification of Si_3_N_4_ Surface

Ammonium hydroxide, hydrogen peroxide, absolute ethanol, APTES, GA, glycine, and sodium cyanoborohydride (NaBH_3_CN) were purchased from Sigma-Aldrich (St. Louis, MO, USA). Small pieces of Si_3_N_4_ wafers (approximately 1 cm × 0.5 cm and 2 cm × 2 cm) and ISFET devices were supplied by the Thai Microelectronic Center (TMEC). These substrates were cleaned in 5% alkaline detergent (Hellmanex III, Hellma GmbH and Co. KG, Germany) using a mechanical ultrasonic cleaner (UCE ultrasonic co., Ltd., Beijing, China) at RT for 15 min to remove dirt particles and other organic contaminants on the surface. Then, silicon substrates were cleaned and activated with RCA-1 solution (NH_4_OH: H_2_O_2_: H_2_O—1: 1: 5) at 65 °C for 30 min. The substrates were thoroughly washed with de-ionized (DI) water and dried with nitrogen gas. In the process, the RCA-1 solution increased the hydrophilicity of the surfaces covered by a native oxide and high-density –OH groups. Before the silanization step, the substrates were dried in the oven at 110 °C for 1 h to complete dehydration, which is critical to prevent APTES bonding to the residual water molecules on the surfaces. The silanization process was carried out using APTES in 90% ethanol at 1% (*v*/*v*) at RT for 1 h. The silicon substrates were finally cured in an oven at 110 °C for 1 h to remove water molecules and form siloxane (Si-O-Si) bonding. Subsequently, the APTES-modified surfaces were functionalized into GA linker (GA, 2.5% *v*/*v* in 10 mM phosphate buffer saline (PBS), pH 8.2) to form aldehyde functional groups on the modified surfaces and immobilize biomolecules such as proteins or antibodies. After that, the GA functionalized surfaces were carefully washed with 10 mM PBS, pH 7.4, dried with nitrogen gas, and used for protein immobilization in the next step. The surface modification, protein immobilization, and antibody detection processes are shown in Figure 10.

### 3.2. Characterization of Modified Si_3_N_4_ Surface

The characterization of the modified Si_3_N_4_ surface was performed using CA analysis, ellipsometry, and AFM to find the degree of wetting, surface thickness, and surface roughness in each modification step, respectively.

#### 3.2.1. Contact Angle (CA) Measurement

The OCA 15EC Optical Contact Angle Measuring Instrument (DataPhysics Instruments GmbH, Regensburg, Germany) was used to determine the surface tension and wettability of the modified Si_3_N_4_ surfaces. The CA was measured in static mode at RT using DI water drops in each step of surface modification. The images were captured at 5 s after 5 µL of the DI water drop touched the surface, and the drop profile was analyzed using drop shape analysis software. The CA values were reported as an average from the measurements of three different areas on the same substrate in triplicate.

#### 3.2.2. Ellipsometry

The layer thicknesses of the modified and functionalized Si_3_N_4_ surfaces in the silanization and cross-linking steps were measured using a VASE2000 spectroscopic ellipsometer (J.A. Woollam, Inc, Lincoln, NE, USA.). The measurements were performed at an incidence angle of 75° in the wavelength range of 250–1650 nm at 5 nm intervals. The WVASE32^®®^ Software program (J.A. Woollam Company (JAWCo), NE, USA.) was used for data analysis.

#### 3.2.3. Atomic Force Microscopy (AFM)

The surface topography of the modified Si_3_N_4_ wafer was examined in tapping mode using a Seiko SPA400 atomic force microscope (Seiko Instruments, Inc., Chiba, Japan) with high-quality silicon tip cantilevers (Applied NanoStructures, Inc., Mountain View, CA, USA). The measurement was taken under an ambient condition with a scanning area of 5 µm × 5 µm and a scan speed of 1 Hz. The surface roughness was defined by the root mean square (RMS) value, which was calculated from the fluctuations in surface height at around the average height in the analyzed section [26]. 

### 3.3. Antigen Immobilization and Antibody Binding

To confirm the existence of the antigen-antibody complex on the modified silicon substrates, fluorescence detection and electrical measurement (gate potential changes) were performed.

#### 3.3.1. Fluorescence Detection

The Si_3_N_4_ wafers were modified and functionalized using the procedures mentioned earlier. The recombinant HLA-A protein (P01) (H00003105-P01, Abnova™, Taipei, Taiwan) and household MICA*010 protein were used as the representatives of HLA and MICA proteins, respectively. The concentrations of 0, 1, 2.5, 5, 10, and 20 µg/mL for each protein were separately immobilized on the modified silicon substrates, whereas 10 µg/mL of HLA-A-purified Max-Pab mouse polyclonal antibody (B01P) (H00003105-B01P, Abnova™) and WJ-1, household MICA antibody were used as the corresponding antibodies. Subsequently, they were immersed in the light-shielded microtubes containing 10 µg/mL of FITC-conjugated goat anti-mouse IgG (clone poly4053, Biolegend, CA, USA) for 1 h to demonstrate the antigen-antibody complex on the Si_3_N_4_ surface using fluorescence signal. Silicon substrates were rinsed three times with a washing solution to remove unbound conjugated antibodies. The fluorescence detection was achieved using a bright field and fluorescence inverted microscope (Model: Olympus BX63, Tokyo, Japan), which was connected to the high-performance digital camera DP73 (Olympus Corporation, Tokyo, Japan). All the images were captured in a fluorescence mode with excitation at 495 nm and emission at 519 nm and an exposure time of 10 s. The purified mouse IgG1, ĸappa-isotype control antibody (clone MOPC-21, Biolegend, CA, USA) was used as a negative control to distinguish the non-specific binding signal from a specific binding signal. Finally, fluorescent signals were analyzed using the ImageJ software and presented as the integrated fluorescence densities.

#### 3.3.2. ISFET Measurement

The detection performance of ISFETs was investigated by electrical measurement in five different modules (bare, cross-linking, protein immobilization, baseline, and antibody binding). The HLA and MICA proteins were prepared as 10 µg/mL and spread on the APTES-GA-modified ISFET at RT for 1 h. A commercial Ag/AgCl reference electrode (Winsense Co., Ltd., Bangkok, Thailand) and the source and drain terminals of the ISFET devices were connected to the readout circuit. A light-shielding type bottle consisting of 10 µM PBS without calcium chloride or magnesium chloride, pH 7.4, was used to avoid the surrounding lights, which cause electrical interference during measurements. To evaluate the optimal constant current and electrical characteristics, the I_ds_-V_gs_ curves of the ISFETs were measured in 10 µM PBS, pH 7.4 at various currents, ranging from 5 µA to 150 µA using a high-precision I_ds_-V_gs_ characterization semiconductor analyzer, B1500A. Subsequently, the graph was plotted between I_ds_ and V_gs_ to establish the optimal range of I_ds_, which provides the linear and parallel trend of the V_gs_. In addition, the gate potential changes were observed every second for 6 min by a multimeter (Agilent U1232A) connected to data logger software. Firstly, the gate voltages (V_gs_) were measured before (V_gs_
_(GA__)_) and after protein immobilization (V_gs_
_(protein__)_) in 10 µM PBS, pH 7.4. The gate potential changes (ΔV_gs_) were monitored at 1 and 6 min as delta V_gs_ (ΔV_gs_ = V_gs_
_(protein__)_ − V_gs_
_(GA__)_). To block the free aldehyde group, the ISFET devices were immersed in 1M glycine solution for 1 h, followed by 150 mM of NaBH_3_CN for 1 h. The excess solution was washed using a washing solution. Then, all ISFETs were immersed in 10 mM PBS, pH 7.4 for 1 h, and the gate potential was measured as V_gs_
_(baseline__)_. After that, the anti-HLA-A, anti-MICA, and mouse IgG isotype control were tested using the antigen-immobilized ISFETs for 1 h. The binding of antibodies was observed by the changes in the gate potential (ΔV_gs_ = V_gs_
_(antibody__)_ − V_gs_
_(baseline__)_). The V_gs_ was collected at 6 min after a constant current (30 µA) was applied to the system. All the electrical measurements were performed at RT. 

### 3.4. Protein Concentration Optimization

The commercial HLA protein solution was prepared in a PBS buffer with concentrations varying from 5 to 15 µg/mL. The household MICA*010 protein solution was also prepared separately, with concentrations between 6.25 and 100 µg/mL. The APTES-GA functionalized surface was immersed in the different concentrations of proteins for 1 h, at RT. The excess protein solution was removed by washing in PBS buffer, pH 7.4. The V_gs_ was calculated after protein immobilization. The gate potential changes after protein immobilization were calculated by ΔV_gs_ = V_gs (protein)_ − V_gs (GA)_.

### 3.5. Method Validations

#### 3.5.1. Dose-Response Curve

The dose-response curve of the developed ISFET biosensor for the detection of anti-HLA and -MICA antibodies was established under optimal conditions. A total of 10 µg/mL of commercial HLA protein and 50 µg/mL of household-MICA*010 proteins were used in the immobilization step. The different concentrations of commercial anti-HLA and anti-MICA (WJ-1) antibodies (0, 5, 10, 20, 40, 80, and 160 µg/mL) were employed on the protein-immobilized ISFET for 1 h, at RT. The purified mouse IgG1, ĸappa-isotype antibody was used as a negative control. The linearity and the slope of the graph were used to identify the detection range and sensitivity of the developed ISFET biosensor, respectively. 

#### 3.5.2. Limit of Detection (LoD) and Limit of Quantitation (LoQ) 

The LoD of an individual analytical procedure is the lowest concentration of the target analyte that can be determined and reliably distinguished from background noise signals. LOQ is the lowest concentration of analyte that can be quantitatively determined with an acceptable level of precision. The LoD and LoQ are regularly estimated using the expression [27]: LoD = 3σ/S(1)
LoQ = 10σ/S(2)
where σ is the standard deviation of the blank (peak voltage at 0 µg/mL of the anti-HLA and -MICA antibody concentration, obtained from at least ten individual blank samples (see Appendix A)) and S is the slope of the calibration curve. 

#### 3.5.3. Cut-Off Determination

The optimal cut-off value of the developed biosensor was calculated from the ΔV_gs_ obtained from the known independent negative sample (the isotype control), which was tested with the positive sample (specific monoclonal antibody) under the same conditions using the formula
Cut-off = α · *X^−^* + f · SD(3)
where α = 1 and f = 3, (i.e., cut-off = mean + 3 times the standard deviation of the known negative samples), SD = a standard deviation obtained from the known negative samples (the isotype antibody in several concentrations, varying from 0–160 µg/mL; see Appendix A).

#### 3.5.4. Analytical Specificity 

The specificity of the ISFET-based biosensor for the detection of anti-HLA and anti-MICA antibodies was evaluated by introducing the irrelevant antibody (mouse IgG1, kappa isotype control) in parallel with the specific antibody at different concentrations, including 0, 5, 10, 20, 40, 80 and 160 µg/mL.

#### 3.5.5. Analytical Precision

The commercial HLA-A protein and household MICA*010 protein were separately immobilized on the APTES-GA-functionalized surface. The reproducibility of the developed biosensor was determined using ten different protein-immobilized ISFET sensors from three different batches against 20 µg/mL of both anti-HLA and -MICA antibodies. The ΔV_gs_ of each ISFET were measured and converted to the protein concentration, using the calibration curve to investigate the between-run coefficient of variation (%CV). 

#### 3.5.6. Sensitivity and Specificity for the Experimental Setting

The sensitivity and specificity of the ISFET-based biosensor were evaluated for anti-HLA and -MICA detection. For the experimental setting, the commercial anti-HLA-A and household anti-MICA (WJ-1) solutions were used to represent the known positive sample. The purified mouse IgG1, ĸappa-isotype control antibody was used as a known negative antibody. The sensitivity and specificity were calculated using a 2 × 2 table. The positive signal was identified using the calculated cut-off value obtained from the study. 

## 4. Conclusions

The presence of antibodies against HLA and MICA proteins is crucial in graft rejection, especially kidney transplantation. An ISFET-based immunosensor was developed as a promising tool, which provides noticeable benefits in terms of portable and miniaturized cost-effectiveness for antibody detection. Silanization with the 1% APTES for 1 h and functionalization with 2.5% GA builds a thin functional group layer for protein immobilization on the Si_3_N_4_ surface. The potential gate changes showed that monoclonal antibodies against HLA and MICA can specifically bind to the corresponding proteins immobilized on the Si_3_N_4_ surface without noise disturbance. Although there is limited evidence for antibody detection using the Si_3_N_4_-ISFET platform, the other analytical performances of the proposed immunosensor were satisfactory and corresponded to the other published studies. In summary, our findings demonstrated a convenient approach to establishing a prototype of the ISFET sensor with the capacity to develop an alternative immunobiosensor for anti-HLA and -MICA antibody detection in future clinical samples.

## Figures and Tables

**Figure 1 molecules-27-06697-f001:**
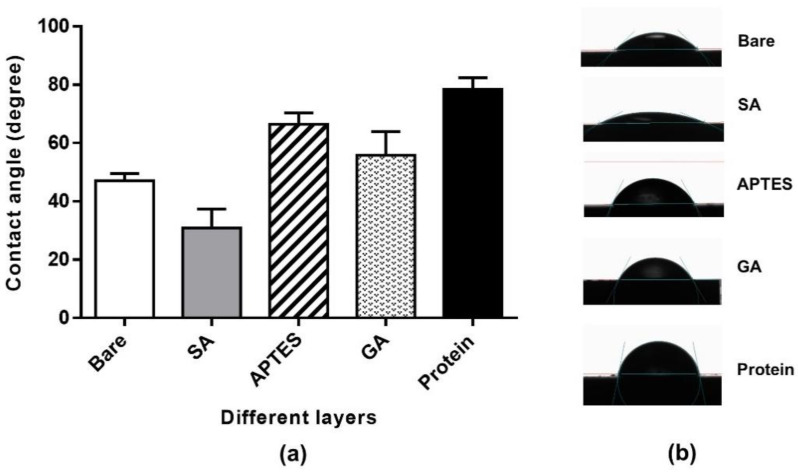
The measurements of static CA on Si_3_N_4_ surface in each step of surface modification and antigen immobilization process. (**a**) Bare; the native Si_3_N_4_ surface without any treatment, SA; surface activation, APTES; Si_3_N_4_ surface after modification with 1% APTES for 1 h, GA; Si_3_N_4_ surface after modification with 2.5% GA for 1 h. Protein; the surface after 20 µg/mL of HLA-A protein immobilization. (**b**) The illustration of the water droplets on the different silicon nitride substrates in each step of surface modification.

**Figure 2 molecules-27-06697-f002:**
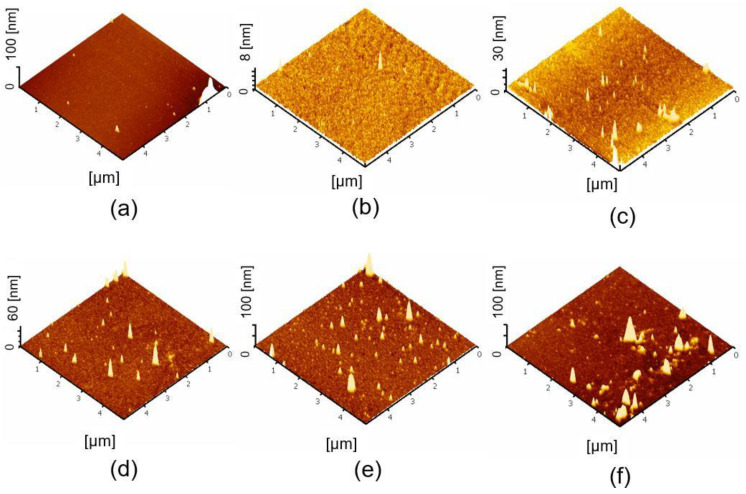
The surface topography analyzed by tapping mode AFM. (**a**) Bare Si_3_N_4_ surface, (**b**) RCA-1-treated Si_3_N_4_ surface, (**c**) APTES-modified Si_3_N_4_ surface, (**d**) APTES-GA-modified surface, (**e**) 20 µg/mL HLA-A-immobilized surface and (**f**) 20 µg/mL MICA*010-immobilized surface.

**Figure 3 molecules-27-06697-f003:**
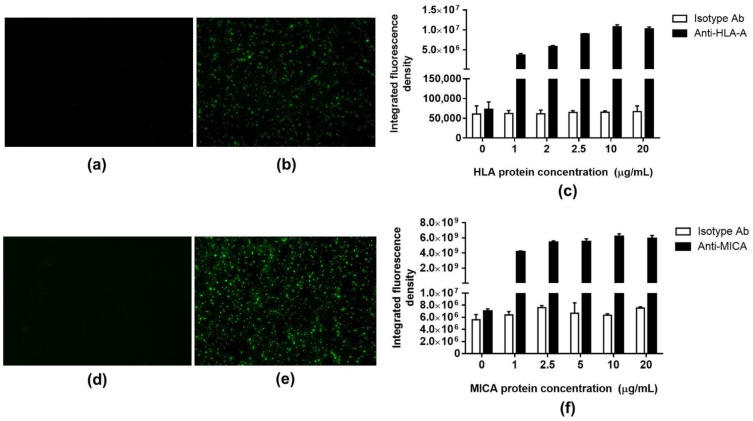
Integrated fluorescence density of FITC-tagged antibodies bound on the silicon surface after immunoassay. (**a**,**d**) the fluorescent signal captured from the silicon wafer probed with IgG1, ĸappa isotype antibody, (**b**) the fluorescent signal captured from anti-HLA probing surface, (**c**) integrated densities of immobilized HLA-A surface probed with anti-HLA-A, (**e**) the fluorescent signal captured from the anti-MICA probing surface, (**f**) integrated densities of immobilized MICA*010 surface probed with anti-MICA.

**Figure 4 molecules-27-06697-f004:**
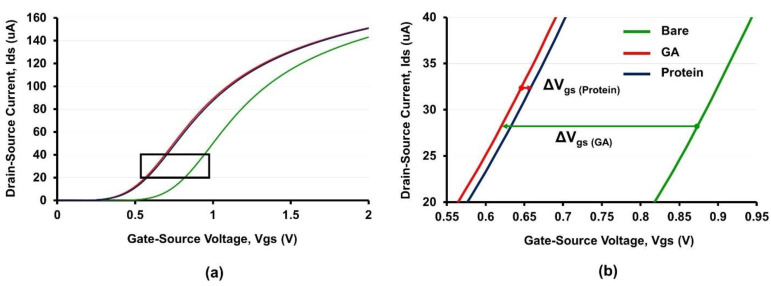
I_ds_-V_gs_ curves after surface functionalization, protein immobilization, and antibody binding. (**a**) Measured gate potential changes against various Ids. (**b**) The extraction of gate potential changes from the linear region of I_ds_-V_gs_ curves.

**Figure 5 molecules-27-06697-f005:**
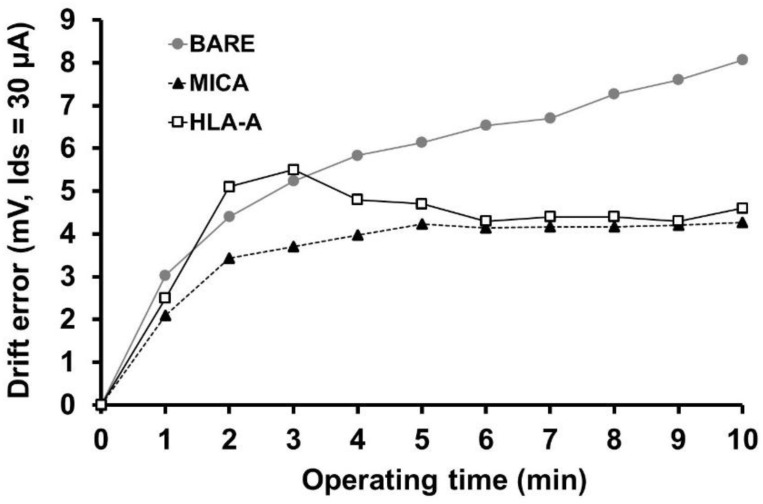
The voltage drift error (V_df_) of ISFETs measured at 1 min intervals up to the 10 min mark in 10 µM PBS, pH 7.4. BARE; untreated Si_3_N_4_ surface, HLA-A; Si_3_N_4_ surface-immobilized with HLA-A protein, MICA; Si_3_N_4_ surface-immobilized with MICA protein.

**Figure 6 molecules-27-06697-f006:**
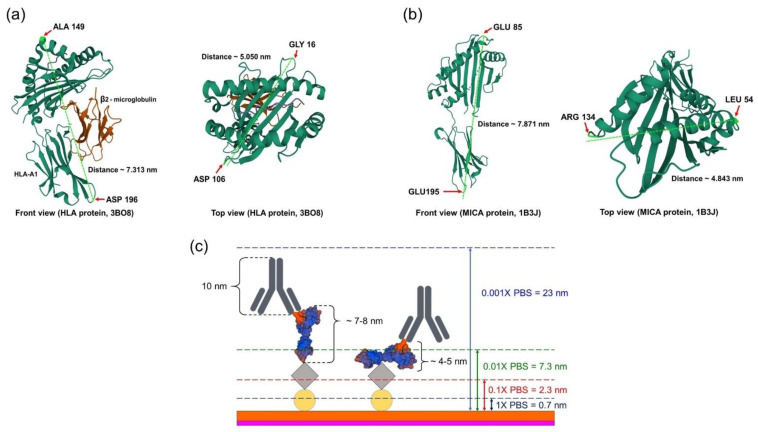
The schematic represents the size of proteins and the length of λ_D_ in different concentrations of PBS at room temperature. (**a**) The size of HLA proteins (PDB code: 3BO8); the longest dimension was determined as between 149 (ALA) and 196 (ASP), while the widest dimension was determined as between 16 (GLY) and 106 (ASP). (**b**) The size of MICA proteins (PDB code: 1B3J); the longest dimension was determined as between 85 (GLU) and 195 (GLU), while the widest dimension was determined as between 54 (LEU) and 134 (ARG). (**c**) The total length of the functionalized surface and immunocomplex on the Si_3_N_4_ substrate was 19–22 nm, compared to the different λ_D__,_ which depends on the PBS concentrations.

**Figure 7 molecules-27-06697-f007:**
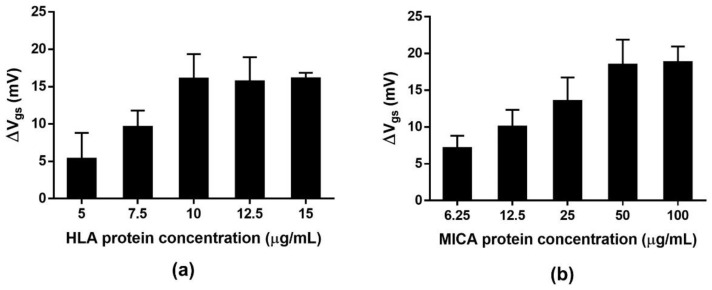
Optimization of HLA and MICA protein immobilization for electrical measurement. (**a**) The ΔV_gs_ obtained various concentrations of HLA protein immobilization. (**b**) The ΔV_gs_ was obtained from various concentrations of MICA protein immobilization.

**Figure 8 molecules-27-06697-f008:**
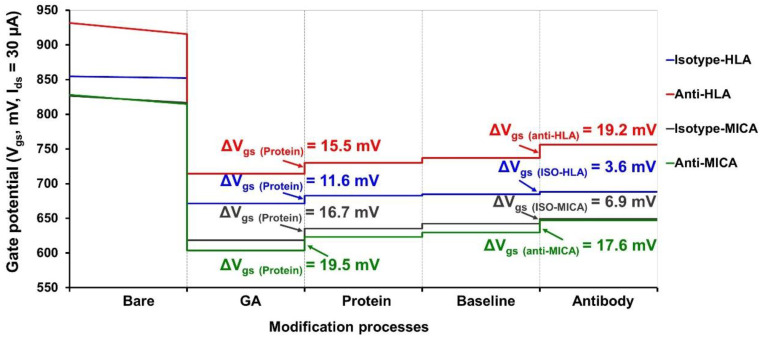
The surface potential measured in each module of the surface modification. The electrical monitoring was performed in five modules, including bare Si_3_N_4_ surface, APTES-GA functionalized surface, protein-immobilized surface, baseline, and antibody-binding. The experiment was carried out by using the optimal protein concentration of HLA and MICA proteins with 40 µg/mL of each antibody.

**Figure 9 molecules-27-06697-f009:**
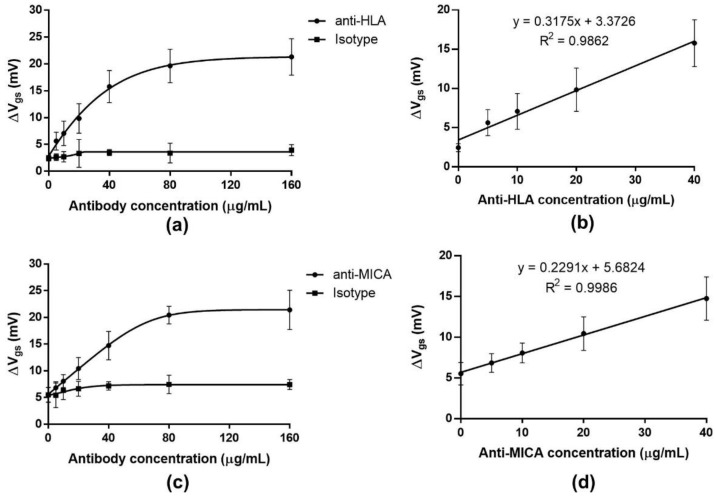
Dose-response curve exhibits the gate potential changes against different concentrate ions of antibodies. (**a**,**c**) represent the dose-response curve obtained from the anti-HLA and -MICA, respectively. (**b**,**d**) represent the linearity range and the slope of the dose-response curve obtained from HLA and MICA antibodies, respectively. Each dataset shows the mean and SD of the ΔV_gs_ calculated from three different ISFETs.

**Figure 10 molecules-27-06697-f010:**
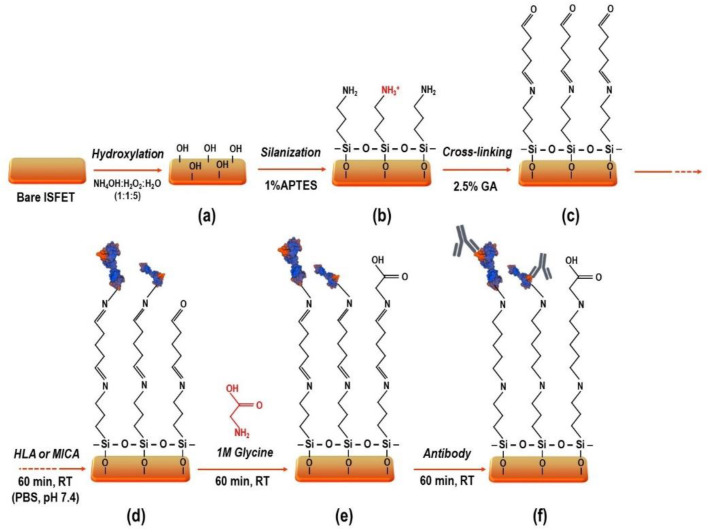
The schematic illustrates the processes for surface modification, protein immobilization, and antibody detection. (**a**) The Si_3_N_4_ substrates’ cleaning using RCA-1 solution; (**b**) the development of silane layer using 1% APTES; (**c**) the formation of covalent bonding using 2.5% GA; (**d**) the immobilization of proteins; (**e**) the blocking of unspecific binding sites using 1M Glycine; (**f**) the binding of antibodies.

**Table 1 molecules-27-06697-t001:** The measurement of the thicknesses of different layers of Si_3_N_4_ modification using APTES and APTES-GA treatment.

Layer	Thickness (nm)	Average	SD
SiO_2_	102.09	102.32	0.70
	103.11		
	101.75		
Si_3_N_4_	198.72	198.59	0.89
	197.65		
	199.41		
APTES	2.57	2.51	0.17
	2.32		
	2.65		
GA	1.15	1.52	0.46
	1.38		
	2.03		

**Table 2 molecules-27-06697-t002:** The reproducibility of the ISFET-based biosensor for anti-HLA and -MICA antibodies detection.

Antibodies (20 µg/mL)	ΔV_gs (antibody)_		Protein Concentration from the Inter-Assay	
	Mean ± SD (mV)	%CV	Mean ± SD (µg/mL)	%CV
Anti-HLA	10.03 ± 0.71	7.09	20.97 ± 2.24	10.69
Anti-MICA	10.31 ± 0.41	4.00	20.21 ± 1.80	8.92

## Data Availability

Research data would be available upon request.

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
