# Peer review of "Biosensors Based on Ion-Sensitive Field-Effect Transistors for HLA and MICA Antibody Detection in Kidney Transplantation"

_molecules, 2022, doi:10.3390/molecules27196697_

Round 1

Reviewer 1 Report (Previous Reviewer 3)

The authors have made a significant modification to the manuscript. The quality of the manuscript is much improved now. Only one minor suggestion, the present title is: Fabrication of Ion-Sensitive Field Effect Transistor-based Biosensors for HLA and MICA Antibodies’ Detection in Kidney Transplantation. Since the focus here is not fabricating the ISFET (considering that no fabrication of the device is performed and the surface functionalization with APTES and GA is very very very common), the authors might consider a title with many highlights on the application, to eliminate misleading.

Author Response

Reviewer 1

Comments and Suggestions for Authors

The authors have made a significant modification to the manuscript. The quality of the manuscript is much improved now. Only one minor suggestion, the present title is: Fabrication of Ion-Sensitive Field Effect Transistor-based Biosensors for HLA and MICA Antibodies’ Detection in Kidney Transplantation. Since the focus here is not fabricating the ISFET (considering that no fabrication of the device is performed and the surface functionalization with APTES and GA is very very very common), the authors might consider a title with many highlights on the application, to eliminate misleading.

Responses:

Thank you so much for the suggestion. We agree with the reviewer. Thus, we have modified the title to “Biosensors Based on Ion-Sensitive Field Effect Transistor for HLA and MICA Antibodies’ Detection in Kidney Transplantation”

We most appreciate the reviews’ efforts to improve the manuscript to the current suitable form for publication.

Reviewer 2 Report (Previous Reviewer 2)

This is an interesting study. I appreciate the efforts which the authors made to respond to reviewers' suggestions and in general, I am satisfied with their response.

Author Response

Reviewer 2

Comments and Suggestions for Authors

This is an interesting study. I appreciate the efforts which the authors made to respond to reviewers' suggestions and in general, I am satisfied with their response.

Responses:

Thank you so much. We most appreciate the reviews’ efforts to improve the manuscript to the current suitable form for publication.

This manuscript is a resubmission of an earlier submission. The following is a list of the peer review reports and author responses from that submission.

Round 1

Reviewer 1 Report

  • This manuscript describes surface functionalization of Si3N4 that is the ISFET sensing material, and HLA and MICA antibodies detection using the functionalized ISFETs.
  • Please indicate the official name of the MHC on the three lines of the abstract, not the abbreviation.
  • In section 2.3.2, sodium cyanoborohydride (NaBH3CN) was used as a component of a blocking reagent. Normally, the NaBH3CN is used to activate Schiff bases. It means that the NaBH3CN is also useful to increase the amount of the immobilized protein. Please mention why the author did not use NaBH3CN during the protein immobilization in the manuscript, since the author claim novelty of surface functionalization process.
  • In Figure 8, it seems better to show the detection limit and detection range to discuss possibility of future practical use of the developed ISFET immnosensor. The author detected the free specific antibodies in this experiment, but HLA and MICA express on cell membranes. The opportunities of contact between the immobilized protein and the protein on the cell surface are expected to differ from the results shown in present experiments.

Reviewer 2 Report

In this study, there were no exciting findings, all of which were reported in previous studies. Although the authors demonstrate the effective surface modification procedure using APTES and GA, these two cross-linking reagents are most commonly used in bio-immobilization, even on ISFET assays. In addition, a similar study which investigated APTES modification for biomolecule immobilization on Si3N4 surface was recently published (doi: 10.1016/j.talanta.2019.120305). The efficiency of surface modification conditions was evaluated by fluorescence microscopy, atomic force microscopy, and ISFET measurement which were also used in this work.  Therefore, in this work, the authors focus on the bio-immobilization of ISFET that is not enough to attract wide attention. I suggest that the authors should focus on the detection and application of anti-HLA and -MICA, which need to present a complete sensing performance in terms of sensitivity, detection limit, specificity, and practical applications. For the current content, it is not suitable for publishing on Molecules unless it has undergone extensive experiments and more substantial content revision.

Reviewer 3 Report

The work does not show adequate novelty in terms of ISFET materials and surface chemistry. The manuscript is immature and must be significantly improved. Please refer to the following details.

  1. English editing is highly needed. Would you please use the service from a reputable English editing provider and get the certification?
  2. The manuscript lacks figural data as valid supporting evidence to complement the bar charts or numerical table. Important missing data are:
  3. Photographs from water contact angle measurement
  4. Figural data from ellipsometry analysis
  5. Images from fluorescence microscope analysis.
  6. The ISFET measurement results are not complete. Step-by-step surface functionalization measurement must be presented, for instance, Bare/APTES/GA/Glycine/probe/target Protein, and the threshold voltage shift plots. Similar data presentation should be done for the isotype detection for specificity study. Not only the bar diagram but please also show the IdVg curve and gate voltage threshold shifting plots.
  1. A series of concentrations of MICA and HLA testing should be performed, and the results should be presented in a calibration plot to show the sensitivity of the sensor and quantify the limit of detection (LoD).
  2. It is highly suggested to make a table of comparisons showing the advantage of the proposed sensor and outcomes over other published works in this area.